# Anisotropic Kondo line defect and ODE/IM correspondence

**Jingxiang Wu**

Perimeter Institute for Theoretical Physics, Waterloo, Ontario, N2L 2Y5, Canada
Dept. of Physics & Astronomy, University of Waterloo, Waterloo, ON N2L 3G1, Canada

## Abstract

We study the anisotropic Kondo line defects in products of chiral $SU(2)$ WZW models. We propose an ODE/IM correspondence for the anisotropic Kondo problems by considering the four-dimensional Chern Simons theory in the trigonometric case. We verify the claim both by explicit perturbative calculations in the ultraviolet and by exact WKB analysis in the infrared. In doing so, we derived the infrared dynamics of a large class of anisotropic line defects.

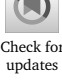

# 1 Introduction

Kondo model was invented to describe a single magnetic impurity in a condensed matter system and now has become a prototypical example of quantum impurity [1–7]. See e.g. [8,9] for a detailed account of the historical developments and further references. In the language of quantum field theory (e.g. [10–14]), a local impurity coupled to the bulk gives rise to a line defect, which is generically nonconformal. In particular, the Kondo line defect that we will consider in this paper is a large class of (nonconformal) chiral line defects where the impurity is only coupled to the chiral half of a bulk conformal field theory. Therefore throughout this paper, we will only be concerned with the chiral half of a CFT since the Kondo defect is transparent to the anti-chiral sector. We will assume a basic familiarity with [15,16] and only review some necessary background to set up the notations.

One of the basic examples is when the bulk CFT is chiral $SU(2)_k$ WZW model with the defect Hamiltonian

$$H = g\, t^a J^a(t,0),\tag{1}$$

where $t^a$ is $n$ dimensional irreducible matrix representation of $SU(2)$ and $J^a$ is the chiral $SU(2)_k$ WZW currents. It produces a line defect,

$$\hat{T}_n[g] := \mathrm{Tr}_{\mathcal{R}_n}\, \mathcal{P} \exp\left( ig \int_0^{2\pi R} d\sigma\ t_a J^a(\sigma,0) \right),\tag{2}$$

with the trace taken in the $n$ dimensional irrep of $SU(2)$. To be well-defined as a quantum operator, one has to carry out a careful renormalization procedure. One regularization scheme is given by [17] where the renormalized operator is explicitly computed in the first few orders of $g$ and $\frac{1}{k}$. (The result for finite $k$ can be found in [15]). Their regularization scheme is particularly nice because it preserves

- translation invariance along the defect so that aside from the renormalization of the coupling $g$, the only local counterterm is the identity

- rigid translation invariance perpendicular to the defect

- global symmetry $SU(2)$

It turns out that the Kondo defect is asymptotically free and dynamically generates a scale $\mu \equiv e^\theta$. Therefore we will denote the Kondo defect below as $\hat{T}_n[\theta]$.

Kondo line defects are not topological. Rather, it is believed to carry the integrable structure of the bulk CFT [18], which, among other things, claims the commutativity for any $n$, $n'$ and $\theta$, $\theta'$

$$\left[\hat{T}_n[\theta], \hat{T}_{n'}[\theta']\right] = 0\,,\tag{3}$$

and the Hirota fusion relation

$$\hat{T}_n\left[\theta + \frac{i\pi}{2}\right]\hat{T}_n\left[\theta - \frac{i\pi}{2}\right] = 1 + \hat{T}_{n-1}[\theta]\hat{T}_{n+1}[\theta]\,.\tag{4}$$

As operator equations, they have been verified explicitly in [15]. It is one of the surprising incarnations of the integrability that (quantum) Kondo line defect corresponds to a (classical) ordinary differential equation in the spirit of ODE/IM correspondence[1] [21]: the expectation value of the Kondo line defect in a state $|\ell\rangle$ is identified with the Stokes data of an ODE

---

[1]See also [19,20] and more references in [16] for lots of further development.

[15,16].[2] The most basic example is the Kondo defect in the chiral $SU(2)_k$ WZW model which corresponds to the ODE[3]

$$\partial_x^2 \psi(x) = \left[ e^{2\theta} e^{2x} (1 + gx)^k + t(x) \right] \psi(x), \tag{5}$$

where $t(x) = 0$ if $|\ell\rangle$ is the vacuum state and for other generic states, $t(x)$ is determined by a set of Bethe equation [19, 20, 25, 26, 28]. See also [29–31] for a more recent discussion. A special case of the ODE (5) in vacuum state at level $k = 1$ has appeared in [32] in the early days of the ODE/IM correspondence.

We think of (5) as the most basic example of ODE/IM correspondence, based on which, one can easily generalize ODE to other chiral algebras associated to $\widehat{su(2)}_k$, including the multichannel $\prod_i SU(2)_{k_i}$ and the coset $\prod_i SU(2)_{k_i}/SU(2)_{\sum_i k_i}$ [15, 16]. Generalizations to higher rank Lie algebras similar to [33] should also be straightforward (e.g. see a discussion for $sl_3$ in [16]).

In general, it is very hard to determine the strongly coupled physics in the infrared where the perturbation theory breaks down. Thanks to the ODE/IM correspondence, one can easily derive the nonperturbative infrared dynamics of a large class of Kondo line defects by utilizing the techniques of exact WKB analysis. See [15, 16] and references therein for more details. Notably, we discovered an interesting pattern of the wall-crossing phenomenon in the complex $\theta$ plane. In particular, as a special case, we reproduced the well-known conjecture of underscreening-overscreening transition from [34, 35].

Let us remark that it is not yet understood if ODE/IM correspondence can be obtained from a direct relationship between two physical systems. We expect one should be able to find its string theory embedding[4] in the same vein as in [36]. As a step towards this end [15], we embed the Kondo problems in the recently proposed four-dimensional Chern Simons theory [37–41] and conjecture an identification of the meromorphic one form with the logarithmic derivative of the potential in (5).

On a different front, in this article, we would like to ask the following natural question: what happens if the coupling between the bulk and impurity spin is anisotropic, i.e. when the defect Hamiltonian is not $SU(2)$ symmetric

$$H = g_\perp \left( t^+ J^-(t, 0) + t^- J^+(t, 0) \right) + 2g_z t^0 J^0(t, 0), \tag{6}$$

with $g_\perp = g_z$ being isotropic. This is a well-studied quantum impurity problem in condensed matter literature starting from [42] and further developed by many others including [43–47]. One can also find extended discussions in reviews like [48–51] and references therein. The perturbation theory yields the renormalization group equation in the following form

$$\beta_{g_z} = C_1 g_\perp^2 + \dots, \tag{7}$$

$$\beta_{g_\perp} = C_2 g_z g_\perp + C_3 g_\perp^2 + \dots, \tag{8}$$

which is symmetric under $g_\perp \longleftrightarrow -g_\perp$. Without loss of generality, let's take $g_\perp > 0$ and the RG flow diagram is shown in 1

Importantly, we have a line of UV fixed point labelled by finite $g_z$ at $g_\perp = 0$ and renormalization will bring us to strong coupling in the infrared. We then expect to have a family

---

[2]Earlier proposal in similar style can also be found in [22].

[3]More precisely, they belong to a mathematical structure called *oper* defined *globally* on Riemann surfaces. Oper was first introduced by A. Beilinson and V. Drinfeld [23]. The term "oper" is motivated by the fact that for most of the classical $G$ one can interpret $G$-opers as differential operators between certain line bundles. One of the relevant facts to us is that $\psi(x)$ transforms as a section of $K^{-1/2}$ between coordinate patches. See a quick introduction for physicists in the appendix of [15] and [16]. More mathematical materials can be found e.g. in [24–27].

[4]One might think of this as an example of the affine generalization of the geometric Langlands program.

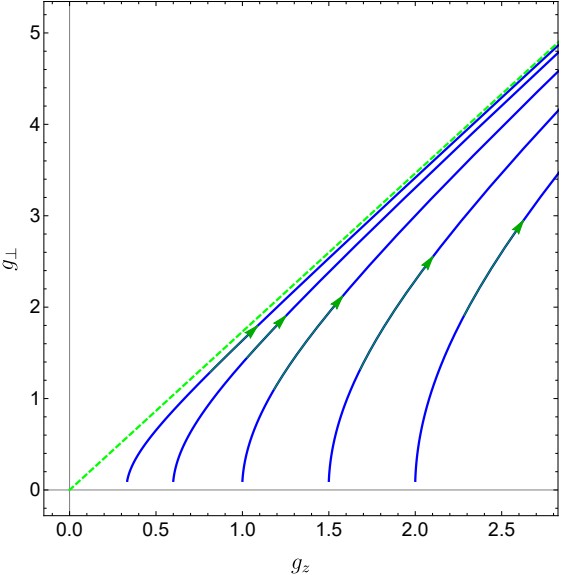

Figure 1: A schematic RG flow diagram for the anisotropic Kondo model (6). The green dashed line is the isotropic flow.

of Kondo line defects labelled by the UV fixed point. In section 2, we will properly define and renormalize the line defect produced by the anisotropic Kondo model. In particular, the quantum line defect we define also make sense for *finite* deformation parameter. We will give explicit results in the leading nontrivial order. In section 3, we will discuss its embedding in the trigonometric 4d Chern Simons theory and propose an ODE (62) that corresponds to the $SU(2)_k$ anisotropic Kondo line defect in the fashion of ODE/IM correspondence. The proposal will be verified both in the UV and IR. In doing so, we derive the infrared dynamics in a variety of classes of line defects. In the simplest case, i.e. physical RG flow of $SU(2)_k$ WZW we reproduce the known result in the literature. In section 4, we generalize the construction to multichannel $\prod_i SU(2)_{k_i}$.

**Note added:** While the manuscript was close to completion, we became aware of [52] which has some overlap with our results. In particular, our proposal of the ODE (62) and (101) is equivalent to (7.7) in [52] via a suitable change of coordinate. As remarked in section 3.1, the perspective developed in this article has pleasantly many different features manifested. We thank Gleb A. Kotousov and Sergei L. Lukyanov for sharing their work with us before publishing.

## 2 Anisotropic Kondo defect

### 2.1 Twist fields and defect changing operators

We are interested in (anisotropic) Kondo defects associated with the $SU(2)_k$ WZW model. Since (anisotropic) Kondo defect will be defined only using chiral currents of the bulk chiral algebra, we can embed $\widehat{su(2)}_k$ into $\left(\widehat{su(2)}_1\right)^{\oplus k}$, which is the chiral half of $k$ compact bosons at the self-dual radius.

Let us start by reviewing some necessary facts about a single compact boson. We will work with the normalization such that[5]

$$\langle \varphi(z)\varphi(w)\rangle = -\frac{1}{2}\log(z-w).$$
(9)

The vertex operator $V_\alpha =: e^{i\alpha\varphi}:$ has conformal dimension $h = \frac{\alpha^2}{4}$ and has the OPE with the current

$$i\partial\varphi(z)V_\alpha(w) \sim \frac{\alpha}{2}\frac{V_\alpha(w)}{z-w}.$$
(10)

In particular, consider the vertex operator $J^\pm \equiv V_{\pm 2}$, they have conformal dimension 1 and the OPE with the current $i\partial\varphi$ as follows

$$i\partial\varphi(z)i\partial\varphi(w) \sim \frac{1/2}{(z-w)^2},$$
(11)

$$i\partial\varphi(z)V_{\pm 2}(w) \sim \pm\frac{V_{\pm 2}}{z-w}.$$
(12)

Therefore the current algebra from $i\partial\varphi$ can be extended by $J^\pm$ to give us the chiral algebra $\widehat{su(2)}_1$.

The current $J^0 \equiv i\partial\varphi$ alone generates a $U(1)$ symmetry with the topological line operator (with anticlockwise orientation)

$$\mathcal{L}_\delta = \exp\delta\oint dz\, i\partial\varphi.$$
(13)

Its actions can be determined from the OPE to be

$$\mathcal{L}_\delta \cdot \varphi = \varphi + \pi\delta,$$
(14)

$$\mathcal{L}_\delta \cdot V_\beta = \exp(\pi i\delta\beta)V_\beta,$$
(15)

so that $\mathcal{L}_1$ is identified with the identity line $\mathcal{L}_0$. The topological line $\mathcal{L}_\delta$ can end on twist fields, which are by definition elements of the defect Hilbert space. For example, the ground state of the defect Hilbert space can be written in terms of vertex operator $V_{\pm\delta}$. If we bring a bulk local operator $J(z)$ around the twist field, it undergoes a group action, as shown pictorially in Fig. 2.

$$\mathcal{L}_\delta \cdot J(z) = J\left(e^{-2\pi i}z\right).$$
(16)

As a result, the current $J^0$ is single-valued so the mode expansion is as usual

$$J^0(z) = \sum_{n\in\mathbb{Z}}\frac{J^0_n}{z^{n+1}},$$
(17)

whereas $J^\pm$ are not

$$e^{\pm 2\pi i\alpha}J^\pm(z) = \mathcal{L}_\alpha \cdot J^\pm(z) = J^\pm\left(e^{-2\pi i}z\right).$$
(18)

Equivalently, the current $J^\pm$ have the following mode expansion

$$J^\pm(z) = \sum_{n\in\mathbb{Z}}\frac{J^\pm_{n\pm\alpha}}{z^{n\pm\alpha}},$$
(19)

where $J^\pm_{n\pm\alpha}$ and $J^0_n$ generate the *twisted* affine algebra, related to untwisted $\widehat{su(2)}_1$ by the spectral flow transformation $U_\alpha$. Please refer to appendix B for more details. Importantly the

---

[5]It is the chiral part of a compact boson at self-dual radius, which in our normalization is $R = 1$.

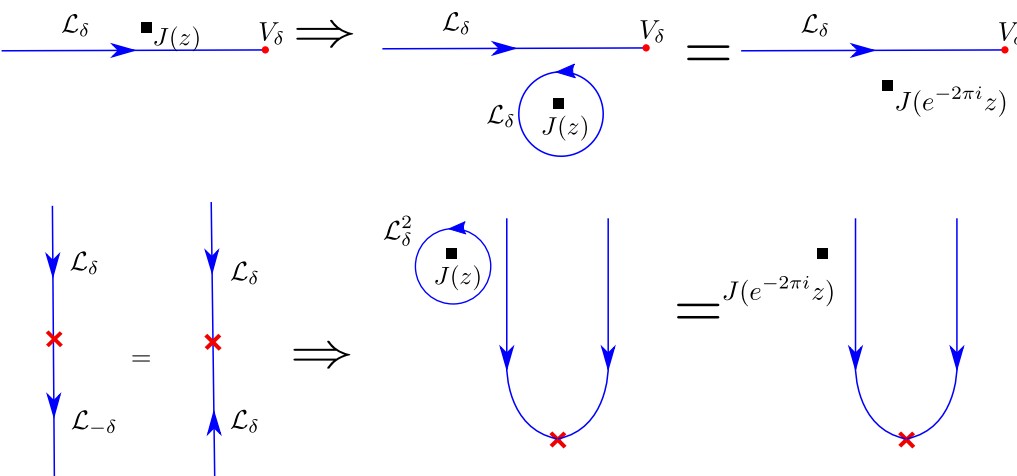

Figure 2: Red dots denote twist fields living at the end of the topological line defect and red crosses denote the defect-changing operators living at the junction of two topological line defects.

Hilbert space after the spectral flow transformation is isomorphic to the untwisted one with a shift of the $L_0$ eigenvalue and $J_0^0$ eigenvalue [53]

$$L_0 \mapsto L_0 + \alpha J_0^0 + \frac{1}{4}\alpha^2, \quad J_0^0 \mapsto J_0^0 + \frac{k}{2}\alpha. \tag{20}$$

In particular, vacuum operator, under the spectral flow $U_\alpha$, maps to an operator with dimension $\frac{\alpha^2}{4}$ living at the end of the twist defect $\mathcal{L}_\alpha$, namely

$$U_\alpha : \mathbb{1} \mapsto V_\alpha. \tag{21}$$

Defect fields are local operators living on the topological line $\mathcal{L}_\delta$. More generally, there are fields, referred to as defect-changing operators, living on the junction of two topological lines, denoted as the red cross in Fig. 2. Similar to the space of twist fields living at the end of the line, the space of defect-changing operators is also isomorphic to the space of bulk local operators related to the spectral flow. To see this, take a look at figure 2. If one brings a local operator $J(z)$ around the defect-changing operator, it gets acted on twice

$$e^{\pm 2\pi i 2\delta} J^\pm(z) = \mathcal{L}_\delta^2 \cdot J^\pm(z) = J^\pm\left(e^{-2\pi i}z\right). \tag{22}$$

Therefore, the space of defect changing operator $\mathcal{H}_{\mathcal{L}_\delta \to \mathcal{L}_{-\delta}}$ living at the junction from a line $\mathcal{L}_\delta$ to a line $\mathcal{L}_{-\delta}$ is simply $U_{2\delta}\mathcal{H}_{\mathrm{bulk}}$. In particular, the image of bulk local current $J^\pm$ under the spectral flow is

$$U_{2\delta} : J^\pm \mapsto V_{\pm 2(1\pm\delta)}. \tag{23}$$

In particular we are interested in $U_{2\delta}\left[J^-\right] = V_{-\beta} \in \mathcal{H}_{\mathcal{L}_\delta \to \mathcal{L}_{-\delta}}$ and $U_{-2\delta}\left[J^+\right] = V_\beta \in \mathcal{H}_{\mathcal{L}_{-\delta} \to \mathcal{L}_\delta}$ with $\beta \equiv 2(1-\delta)$ which have conformal dimension $h = \frac{\beta^2}{4} < 1$ if $\delta \in (0,2)$.

## 2.2 Ultraviolet analysis of the anisotropic Kondo defects

We are now ready to give a precise definition of the anisotropic Kondo defects. In particular, the definition makes sense even for finite deformation parameter $\delta$. We will consider the case

of spin $\frac{1}{2}$, or equivalently $n = 2$. Consideration of higher spin will be left to future work, as we will explain in section 4.

Consider $k$ copies of the chiral algebra $\widehat{su(2)}_1$ discussed in section 2.1

$$J^{0,i} = i\partial\varphi^i, \quad J^{\pm,i} = e^{\pm i2\varphi^i}, \tag{24}$$

which contains the chiral algebra $\widehat{su(2)}_k$ generated by the total currents

$$\mathcal{J}^0 = \sum_i J^{0,i}, \quad \mathcal{J}^\pm = \sum_i J^{\pm,i}, \tag{25}$$

such that

$$\mathcal{J}^0(z)\mathcal{J}^0(w) \sim \frac{k/2}{(z-w)^2}, \tag{26}$$

$$\mathcal{J}^0(z)\mathcal{J}^\pm(w) \sim \pm\frac{\mathcal{J}^\pm(w)}{z-w}, \tag{27}$$

$$\mathcal{J}^+(z)\mathcal{J}^-(w) \sim \frac{k}{(z-w)^2} + \frac{2\mathcal{J}^0(w)}{z-w}. \tag{28}$$

Just like $k = 1$, $\mathcal{J}^0$ generates a $U(1)$ global symmetry whose topological symmetry line is

$$\mathcal{L}_\alpha = \exp\alpha \oint dz \sum_i i\partial\varphi^i. \tag{29}$$

The UV fixed point of the spin $\frac{1}{2}$ anisotropic Kondo defect line of anisotropicity $\delta$ is defined to be the direct sum of two such lines $\mathcal{L}_\delta \oplus \mathcal{L}_{-\delta}$, given explicitly by

$$\exp\delta\sigma_z \oint dz \sum_i i\partial\varphi^i = \exp 2\delta \oint dz\; t^0 \mathcal{J}^0(z), \tag{30}$$

with $t^0 \equiv \frac{1}{2}\sigma_z$ is half of the Pauli matrix $\sigma_Z$. In other words, it is a deformation of a direct sum of two identity lines $\mathcal{L}_0 \oplus \mathcal{L}_0$ by the exactly marginal operator $t^0 \mathcal{J}^0$. As we reviewed in the previous section, the space of the defect-changing operators is related to the space of bulk local operators via the spectral flow operation. In particular, we are interested in $U_{2\delta}\left[t^+\mathcal{J}^-\right] \in \mathcal{H}_{\mathcal{L}_\delta \to \mathcal{L}_{-\delta}}$ and $U_{2\delta}\left[t^-\mathcal{J}^+\right] \in \mathcal{H}_{\mathcal{L}_{-\delta} \to \mathcal{L}_\delta}$ that explicitly take the form

$$\left[t^\pm\mathcal{J}^\mp\right]_\delta \equiv t^\mp \sum_{a=1}^k : e^{\pm i\vec{\beta}_a \cdot \vec{\varphi}} :, \tag{31}$$

where $\beta_a$ are vectors of $k$ components

$$\frac{\vec{\beta}_a}{2} = \vec{e}_a - \delta\mathbb{1} = (-\delta, -\delta, \dots, 1-\delta, -\delta, \dots, -\delta). \tag{32}$$

The operator $\left[t^\pm\mathcal{J}^\mp\right]_\delta$ has dimension

$$h = \frac{1}{4}\vec{\beta}_a^{\,2} = k\delta^2 - 2\delta + 1, \tag{33}$$

which is smaller than 1 when $\delta \in [0, \frac{2}{k}]$. Since they are slightly relevant, it makes sense to turn them on and formally write them as

$$\text{Tr}\,\text{Pexp}\left[\oint dz\; \lambda\left[t^-\mathcal{J}^+\right]_\delta + \lambda\left[t^-\mathcal{J}^+\right]_\delta + 2\delta t^0 \mathcal{J}^0\right] \rightsquigarrow \hat{T}_2(\lambda, \delta). \tag{34}$$

This expression is only classical since it subjects to renormalization once $\lambda$ is nonzero. We will refer to the line defect after an appropriate quantization as the *anisotropic Kondo* defect $\hat{T}_2(\lambda, \delta)$, where the subscript 2 comes from the fact the trace is taken in a two-dimensional representation of $su(2)$. It is then one of the tasks of this section to quantize this line defect. When $\delta = \lambda$, we are reduced to the isotropic Kondo defect, where a nice recipe of quantization has been given in [17] and further studied in [15].

We are interested in the expectation value of $\hat{T}_2(\lambda, \delta)$. More generally, given twist parameter $\alpha$, we can also introduce a twist defect line $\mathcal{L}_\alpha$ with twist fields $V_{\pm\alpha} \equiv : e^{i\vec{\alpha}\cdot\vec{\varphi}} :$ living at the ends, where $\vec{\alpha}$ is a $k$-component vector $\alpha\mathbb{1} = (\alpha, \alpha, \ldots, \alpha)$. This is shown pictorially in figure 3. Then the correlation function $\langle V_{-\alpha}(\infty)\hat{T}_2(\lambda, \delta)V_\alpha(0)\rangle$ can be loosely written as

$$\langle\alpha|\hat{T}_2(\lambda, \delta)|\alpha\rangle, \tag{35}$$

interpretted as the expectation value of the Kondo defect $\hat{T}_2(\lambda, \delta)$ in the state $|\alpha\rangle$.

We will work in the renormalization scheme where $\delta$ is not being renormalized At the leading order, we can turn off the coupling $g$ and Kondo defect reads

$$\mathrm{Tr}\, e^{2\pi i\widetilde{\alpha}t^0}, \tag{36}$$

where we insert a twist inside the trace, which is necessary to make sure the integrand makes sense, i.e. single-valued. To figure out what is the correct choice of $\widetilde{\alpha}$, recall that the defect operators $\left[t^\mp\mathcal{J}^\pm\right]_\delta$ are not single-valued around the circle if there is no such insertion. More specifically, as depicted in figure 3 whenever they go cross the line $\mathcal{L}_\alpha$, they pick up a phase $\exp i\pi\vec{\alpha}\cdot\vec{\beta}_a$. To cancel this phase, one can think of inserting $e^{2\pi i\widetilde{\alpha}t^0}$ at the intersection of the Kondo defect and $\mathcal{L}_\alpha$ line, shown as the pink triangle in figure 3. Due to the identity

$$e^{i2\pi\widetilde{\alpha}t^0}\left(\left[t^-\mathcal{J}^+\right]_\delta(z) + U\left[t^+\mathcal{J}^-\right]_\delta(z) + 2t^0\mathcal{J}^0(z)\right)$$
$$= \left(e^{-2\pi i\widetilde{\alpha}}\left[t^-\mathcal{J}^+\right]_\delta(z) + e^{2\pi i\widetilde{\alpha}}\left[t^+\mathcal{J}^-\right]_\delta(z) + 2t^0\mathcal{J}^0(z)\right)e^{i2\pi\widetilde{\alpha}t^0}, \tag{37}$$

we can set $\widetilde{\alpha} = \frac{1}{2}\vec{\alpha}\cdot\vec{\beta}_a$ so that the phase from crossing the line $\mathcal{L}_\alpha$ cancels the phase from commuting with the insertion $e^{2\pi i\widetilde{\alpha}t^0}$. Therefore at the leading order, we find

$$\langle\alpha|\hat{T}_2(\lambda, \delta)|\alpha\rangle = \mathrm{Tr}\, e^{2\pi i\widetilde{\alpha}t^0} + O(\lambda^2) = 2\cos\pi\widetilde{\alpha}. \tag{38}$$

$O(\lambda)$ order has to vanish since one cannot insert a single defect-changing operator on the Kondo line defect. At the order of $O(\lambda^2)$, we have

$$\lambda^2 \oint_{z_1, z_2} \mathrm{Tr}\, e^{2\pi i\widetilde{\alpha}t^0}\langle\alpha|\left[t^+\mathcal{J}^-\right]_\delta(z_1)\left[t^-\mathcal{J}^+\right]_\delta(z_2)|\alpha\rangle, \tag{39}$$

the integrand is simply a correlation function of four vertex operators, which equals

$$e^{\pi i\widetilde{\alpha}}\left(\frac{z_2}{z_1}\right)^{\widetilde{\alpha}}\left(\sum_{i,j}\delta(\vec{\beta}_i + \vec{\beta}_j)z_{12}^{-\frac{1}{2}\vec{\beta}_i\cdot\vec{\beta}_j}\right) = e^{\pi i\widetilde{\alpha}}k\left(\frac{z_2}{z_1}\right)^{\widetilde{\alpha}}z_{12}^{-\frac{1}{2}\widetilde{\beta}^2}, \tag{40}$$

where

$$\vec{\beta}_i \cdot \vec{\beta}_j = 4(k\delta^2 - 2\delta), \quad i \neq j, \tag{41}$$

$$\widetilde{\beta}^2 := \vec{\beta}_i^2 = 4(k\delta^2 - 2\delta + 1), \quad i = 1, \ldots, k, \tag{42}$$

$$\widetilde{\alpha} := \frac{1}{2}\vec{\alpha}\cdot\vec{\beta}_i = \alpha(1 - k\delta), \quad i = 1, \ldots, k. \tag{43}$$

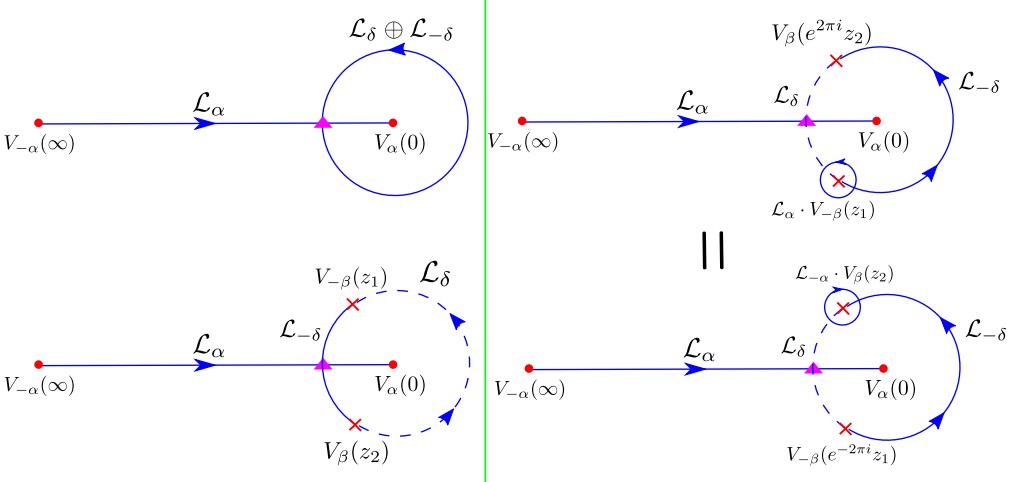

Figure 3: The correlation function $\langle V_{-\alpha}(\infty)\hat{T}_2(\lambda,\delta)V_\alpha(0)\rangle$ involving twist lines $\mathcal{L}_\alpha$ and the anisotropic Kondo lines $\hat{T}_2(\lambda,\delta)$. The top/bottom left diagrams are for the computations at the order $g^0$ and $g^1$ respectively. On the right is a pictorial derivation of the twisting factor inserted on the pink triangle.

The contour integral is over the configuration space $\mathrm{Conf}_2(S^1)$ of two ordered points on the circle of radius $R$. Explicitly, if we parametrize $z_a = e^{i\zeta_a}$ for $a = 1, 2$, then the integration is a two dimensional integral of $\zeta_1$ and $\zeta_2$ over $0 < \zeta_1 < \zeta_2 < 2\pi$ while identifying $0$ and $2\pi$. We will review some basic facts about configuration space in appendix A, where we explain $\mathrm{Conf}_2(S^1)$ is a cylinder $S^1 \times [0, R]$. The integral is greatly simplified if we choose a coordinate system that is rotationally invariant along $S^1$. The result yields

$$\langle \alpha | T(\delta, g) | \alpha \rangle = 2\cos\pi\widetilde{\alpha}$$
$$+ k\lambda^2 e^{\theta(2-\widetilde{\beta}^2/2)} \frac{2^{\widetilde{\beta}^2/2}}{\pi}\left(\cos 2\pi\widetilde{\alpha} - \cos\frac{\pi\widetilde{\beta}^2}{2}\right)\Gamma\left[1 - \frac{\widetilde{\beta}^2}{2}\right]\Gamma\left[\frac{\widetilde{\beta}^2}{4} + \widetilde{\alpha}\right]\Gamma\left[\frac{\widetilde{\beta}^2}{4} - \widetilde{\alpha}\right] \quad (44)$$
$$+ O(\lambda^4),$$

where we renamed the circumference of the cylinder to be $R \equiv e^\theta$. Note that this makes sense since $\lambda$ has a positive dimension and we have a nice expansion in terms of a dimensionless parameter

$$\lambda \exp\left(\theta\left(1 - \frac{\widetilde{\beta}^2}{4}\right)\right). \quad (45)$$

Some remarks are in order. Obviously, the renormalization of the anisotropic Kondo line defect described in this section is much easier than the isotropic case given in [15, 17]. In fact, this is a generic feature of conformal perturbation theory with relevant couplings, where the renormalization is made easier by an analytical continuation of the parameter, i.e. $\delta$ in this case.

Due to the commutativity of the Kondo operators (3), the correlation function (44) are essentially nonlocal integral of motions [54]. In fact, the Kondo defects are expected to coincide with the transfer matrix in [54] after an appropriate identification. We discuss more of this in section 4.

# 3 Anisotropic ODE

## 3.1 Proposal

What should be the corresponding ODE for the anisotropic Kondo defect? Since the ODE for the isotropic Kondo defect is obtained from the four-dimensional Chern Simons theory in the rational setting, it is natural to expect the anisotropic ODE can be found in 4d CS in the trigonometric setting. Let us first recall some key steps in the rational construction [15].

The action for the 4d CS theory [37–39] is

$$\frac{1}{\hbar} \int \omega \, dz \wedge \mathrm{CS}(A), \tag{46}$$

where $\mathrm{CS}(A) \equiv \mathrm{Tr}\left(A dA + \frac{2}{3} A \wedge A \wedge A\right)$ is the Chern Simons three form built out of the partial connection $A = A_x dx + A_y dy + A_{\bar z} d\bar z$. Classically, rational case just corresponds to considering the spacetime to be $\mathbb{R}^2 \times \mathbb{C}$ and $\omega = 1$. It is shown in [15] that if we couple to the 4d CS a 2d chiral WZW model living on a surface defect wrapping $\mathbb{R}^2 \times \{z_0\}$, we obtain an isotropic Kondo problem after integrating out the transverse $z$ direction. The Wilson line wrapping a line $L \subset \mathbb{R}^2 \times \{z\}$ will become a Kondo defect in the two-dimensional system with $z - z_0$ playing the role of the spectral parameter. The commutativity (3) and Hirota relation (4) of Kondo line defect then follow automatically from the properties of the Wilson lines.

There is one important subtlety to the story above we would like to stress. The coupling between the chiral WZW on the surface defect and the bulk gauge field will induce a gauge anomaly. To cancel this anomaly, we need to correct the one form $\omega$ by

$$\frac{dz}{\hbar} \to \frac{dz}{\hbar} + \frac{k}{2} \frac{1}{z - z_0} dz \equiv w(z) dz, \tag{47}$$

so that the *spectral parameter* $\theta$ is identitied to be the primitive $d\theta = \omega(z) dz$. The conjecture proposed in [15] is then the identification between the (twisted) meromorphic one form $\omega(z) dz$ and the logarithmic derivative of the quadratic differential $P(x) dx^2$ from the ODE

$$\omega \longleftrightarrow \frac{1}{2} \frac{\partial P}{P}, \tag{48}$$

from which the ODE takes the form

$$\partial_x^2 \psi(x) = e^{2\theta} P(x) \psi(x). \tag{49}$$

By renaming the coupling $\frac{\hbar}{z_0} \to -g$, we find the isotropic ODE given in [15]:

$$\partial_x^2 \psi(x; \theta) = e^{2\theta} e^{2x} (1 + gx)^k \psi(x; \theta). \tag{50}$$

From rational case to trigonometric case, classically one just needs to replace the one form $dz$ by $\frac{dz}{z}$ and work with $\mathbb{C}^*$ instead of $\mathbb{C}$. On the other hand, since the anomaly is a UV effect which is not sensitive to the global structure, we can therefore, without doing any computations, find the correction due to the gauge anomaly of the same form

$$\frac{1}{\hbar} \frac{dz}{z} \to \frac{1}{\hbar} \frac{dz}{z} \left(1 + \frac{k}{2} \frac{\hbar}{z/z_0 - 1}\right) \equiv dx \left(1 + \frac{k}{2} \frac{\epsilon g}{e^{\epsilon x} - g}\right), \tag{51}$$

where $z = e^{\epsilon x}$, $\epsilon = \hbar$, and $z_0 = g$. In $x$ coordinate, we find the correct residue $\frac{k}{2}$ at the pole $x_0 = \log g$, as expected from (47), which leads to the following proposal

$$\partial_x^2 \psi(x; \theta) = e^{2\theta} e^{2x} \left(1 - g e^{-\epsilon x}\right)^k \psi(x; \theta). \tag{52}$$

In fact, the ODE for the *single-channel* anisotropic Kondo problem for the vacuum state in chiral $SU(2)_k$ WZW has been proposed[6] by S. Lukyanov in [22]

$$\left[ -\partial_u^2 + \kappa^2 \left( e^{-\frac{bu}{Q}} + e^{\frac{u}{bQ}} \right)^k \right] \Theta(u) = 0 \,, \tag{53}$$

where $Q = b + b^{-1}$. After a coordinate transformation $x \equiv \epsilon^{-1}(u + \log(-g))$, $k\epsilon \equiv 2bQ$ and we find exactly (52). As a special case $k = 1$, this is precisely the Generalized Mathieu equation that was proposed by Al. Zamolodchikov in [55] to correspond to Liouville theory.[7] Subsequent studies including exact WKB analysis, including at the self-dual point $b = 1$ can also be found in e.g. [56–59].

Compared to (53), the form of (52) has many new features manifested. Importantly, (52) takes a universal form as (50) since they both have 4d Chern Simons origin. Therefore, as we will show below, we can translate effortlessly most of the techniques we understand well in the isotropic case, including the Hirota equation, WKB analysis and a straightforward generalization to the excited states and the multichannel $\prod_i SU(2)_{k_i}$. Noticeably, as shown in sections below, (52) enables a clear match of physical observables with defect RG flows, which have otherwise remained elusive.

We will refer to $P(x) \equiv e^{2x} \left( 1 - g e^{-\epsilon x} \right)^k$ as the potential and consider

$$0 < \epsilon < \frac{2}{k} \,. \tag{54}$$

With this choice, the Stokes data at infinity is still defined in terms of small solutions along Stokes lines at large positive real part of $x$, spaced by $i\pi$ in the $x$ plane and we can avoid the appearance of Stokes sectors at negative infinity. We will discuss the structure of the WKB diagram in detail later in section 3.3.

The construction we use to extract Stokes data and make contact with the Kondo defect is the same as in the isotropic case [15], which we now review. When the real part of $x + \theta$ is large, the potential is dominated by the part $e^{2\theta} e^{2x}$, where we define *small solutions*. Let's start by defining a small solution $\psi_0(x; \theta)$ to be the unique solution (up to normalization) that decreases asymptotically fast along the line[8] of large real positive $x + \theta$. We fix the normalization of $\psi_0$ so that it agrees with the WKB asymptotics for large positive real $x + \theta$

$$\psi_0(x; \theta) \sim \frac{1}{\sqrt{2e^\theta P(x)}} e^{-e^\theta \int_{-\infty}^x \sqrt{P(y)} dy} \,. \tag{55}$$

Then we define an infinite sequence of small solutions

$$\psi_n(x; \theta) = \psi_0(x; \theta + \pi i n), \quad n \in \mathbb{Z} \,, \tag{56}$$

which can be easily seen to have the asymptotics (55) at large positive real $x + \theta + n\pi i$. This normalization ensures that all the Wronskians[9] between neighboring solutions equals $-i$ identically, i.e. $i(\psi_n, \psi_{n+1}) = 1$. We further define $T$-functions to be

$$T_n(\theta) = i \left( \psi_0 \left( x; \theta - \frac{i\pi n}{2} \right), \psi_0 \left( x; \theta + \frac{i\pi n}{2} \right) \right) \,. \tag{57}$$

The collection of the $T$-functions encodes the Stokes data of the ODE (52).

---

[6]We thank S. Lukyanov for letting us know of his work and the stimulating discussions.
[7]We thank Davide Fioravanti and Marco Rossi for the correspondence.
[8]Here for simplicity, we assume $g$ is real. If we analytically continue $g$, we just need to adjust the imaginary part of $x + \theta$ accordingly to keep on the line where $e^\theta e^x (1 - g e^{-\epsilon x})^{k/2}$.
[9]Given any two functions $f$ and $g$, the Wronskian is defined by $(f, g) = f'g - fg'$.

Compared to the isotropic case in [15], a new feature is that the equation (52) is invariant under $x \to x - \frac{2\pi i}{\epsilon}$ accompanied by $\theta \to \theta + \frac{2\pi i}{\epsilon}$. As we defined above in (55), the small solutions have a behaviour at infinity controlled by a hypergeometric function

$$e^{\theta} \int_{-\infty}^{x} dy \, e^{y} \left(1 - g e^{-\epsilon y}\right)^{k/2}, \tag{58}$$

which is invariant under these translations. We thus have

$$\psi_n \left(x - \frac{2\pi i}{\epsilon}; \theta + \frac{2\pi i}{\epsilon}\right) = \psi_n(x; \theta), \tag{59}$$

and thus the $T$ functions are periodic under $\theta \to \theta + \frac{2\pi i}{\epsilon}$, i.e. to be functions of the spectral parameter $z \equiv e^{\epsilon \theta}$.

Hirota relation [60–62] automatically follow from the construction

$$T_n \left[\theta - i\frac{\pi}{2}\right] T_n \left[\theta + i\frac{\pi}{2}\right] = 1 + T_{n-1}[\theta] T_{n+1}[\theta], \tag{60}$$

which takes the same form as the isotropic $SU(2)$ Hirota relation [16]. However, it is customary to write it in terms of the spectral parameter $z$, where the Hirota relations become multiplicative, involving multiplicative shifts of $z$ by powers of $q = e^{i\pi \epsilon}$.

$$T_n \left[q^{-\frac{1}{2}} z\right] T_n \left[q^{\frac{1}{2}} z\right] = 1 + T_{n-1}[z] T_{n+1}[z]. \tag{61}$$

From the seminal work of [19, 20], given the ODE for the vacuum state, one can find the one for excited states by introducing singularities of trivial monodromy. Since the ODE (52) for the vacuum state takes the same form as the isotropic one (50), the recipe to write down the ODE for excited states is straightforward. One just need to add $t(x) = a_+^2 + \partial_x a_+(x)$ to the potential, so that

$$\partial_x^2 \psi(x) = \left(e^{2\theta} e^{2x} \left(1 - g e^{-\epsilon x}\right)^k + t(x)\right) \psi(x), \tag{62}$$

with

$$a_+(x) = -\alpha_+ - \epsilon l \frac{e^{\epsilon s}}{e^{\epsilon x} - e^{\epsilon s}} + \epsilon \sum_i \frac{e^{\epsilon u_i}}{e^{\epsilon x} - e^{\epsilon u_i}} - \epsilon \sum_i \frac{e^{\epsilon u_i'}}{e^{\epsilon x} - e^{\epsilon u_i'}}, \tag{63}$$

and the requirement that all the singularities $s$, $u_i$ and $u_i'$ have trivial monodromy. This is done by defining $a_-(x) \equiv -a_+(x) - \frac{1}{2} \frac{\partial_x P(x)}{P(x)}$.

$$a_-(x) = -\alpha_- - \epsilon \left(\frac{k}{2} - l\right) \frac{e^{\epsilon s_a}}{e^{\epsilon x} - e^{\epsilon s_a}} + \epsilon \sum_i \frac{e^{\epsilon u_i'}}{e^{\epsilon x} - e^{\epsilon u_i'}} - \epsilon \sum_i \frac{e^{\epsilon u_i}}{e^{\epsilon x} - e^{\epsilon u_i}}. \tag{64}$$

Note that, close to each pole we have the same behaviour as in the isotropic case [16]. For example, close to $s_a$, we have

$$a_+(x) \sim -\frac{l}{x - s_a}. \tag{65}$$

Then by the same reasoning given in [16, 25, 26, 28–30, 63], the trivial monodromy condition is just realized by the Bethe equation

$$-\alpha_+ - \epsilon l \frac{e^{\epsilon s}}{e^{\epsilon u_j} - e^{\epsilon s}} + \epsilon \sum_{j \neq i} \frac{e^{\epsilon u_i}}{e^{\epsilon u_j} - e^{\epsilon u_i}} - \epsilon \sum_i \frac{e^{\epsilon u_i'}}{e^{\epsilon u_j} - e^{\epsilon u_i'}} = 0, \tag{66}$$

$$-\alpha_- - \epsilon \left(\frac{k}{2} - l\right) \frac{e^{\epsilon s_a}}{e^{\epsilon u_j'} - e^{\epsilon s_a}} + \epsilon \sum_{j \neq i} \frac{e^{\epsilon u_i'}}{e^{\epsilon u_j'} - e^{\epsilon u_i'}} - \epsilon \sum_i \frac{e^{\epsilon u_i}}{e^{\epsilon u_j'} - e^{\epsilon u_i}} = 0. \tag{67}$$

In the fashion of ODE/IM correspondence, we claim the following identification [15, 16]

$$\left\langle \ell \left| \hat{T}_n \right| \ell \right\rangle = i \left( \psi_0 \left( x; \theta - \frac{i\pi n}{2} \right), \psi_0 \left( x, \theta + \frac{i\pi n}{2} \right) \right). \tag{68}$$

On the left-hand side is the expectation value of the anisotropic Kondo line defect defined in section 2 in the state $|\ell\rangle$, which, by state-operator correspondence, can be either genuine bulk local operators or twist field living at the end of a twist topological line, as we have discussed in section 2.

Before we conclude this section, let us comment on the isotropic limit. We expect to find the isotropic ODE by taking the limit $\epsilon \to 0$ in an appropriate way. For example, if we expand the potential in small $\epsilon$

$$e^{2\theta} e^{2x} (1 - g e^{-\epsilon x})^k \sim e^{2\theta} e^{2x} (\epsilon g)^k \left( x + \frac{1-g}{\epsilon g} \right)^k. \tag{69}$$

Upon a shift of coordinate, we find $e^{2\theta} (\epsilon g)^k e^{-2\frac{1-g}{\epsilon g}} e^{2x} x^k$. On the other hand, a coordinate shift of $e^{2\hat{\theta}} e^{2x} (1 + \hat{g} x)^k$ yields $e^{2\hat{\theta}} \hat{g}^k e^{-2/\hat{g}} e^{2x} x^k$, we therefore find the identification

$$\hat{g} \leftrightarrow \epsilon g, \quad \hat{\theta} \leftrightarrow \theta + \frac{1}{\epsilon}. \tag{70}$$

The same is true for the Miura part $t(x) = a(x)^2 + \partial_x a(x)$ as well. For example, if

$$a(x) = -\alpha - \frac{\epsilon l e^{\epsilon s}}{e^{\epsilon x} - e^{\epsilon s}}. \tag{71}$$

Expand in the limit $\epsilon \to 0$

$$a(x) \sim -\frac{l}{x-s} + \dots, \tag{72}$$

as expected.

In the following sections, we perform explicit computations both in the UV and in the IR in the fashion of [15] to verify the claim. As the fact that excited states are controlled by the addition of $t(x)$ in (62) follows immediately from the same reasoning in the isotropic case, we will not replicate the same computation in the most general case, rather just to consider $a_+(x) = -\tilde{\alpha}$ and $a_-(x) = 1 + \tilde{\alpha}$, leading to the following ODE

$$\psi'' = \left[ e^{2\theta} e^{2x} (1 - g e^{-\epsilon x})^k + \tilde{\alpha}^2 \right] \psi, \tag{73}$$

## 3.2 Ultraviolet analysis

In this section, we perform the explicit UV analysis, i.e. small $g$ expansion, to verify the claim (68).

The first observation is that if we perform a shift of the coordinate $x \mapsto x + x_0$ in (52), with $g \equiv e^{\epsilon x_0}$, the potential then becomes $e^{2\theta + 2x_0} e^{2x} (1 - e^{-\epsilon x})^k$, which depends on three independent parameters $k$, $\epsilon$ and

$$g_{\text{eff}}(\theta) \equiv \left( e^{2\theta + 2x_0} \right)^{\epsilon/2} = g e^{\epsilon \theta}. \tag{74}$$

We work with the convention that the physical RG flow corresponds to increasing $\theta$ along the real axis while keeping $\epsilon$ and $k$ constant.

Then it is not hard to see we have all the ingredients to match the UV physics of the anisotropic Kondo problem:

1. $k$ should correspond to the level of the bulk chiral $SU(2)_k$ WZW model.

2. The RG flows of the Kondo defect are labelled by UV fixed points, defined by a $\delta t_3 J^3$ deformation. This role is played by parameter $\epsilon$.

3. $e^\theta$ corresponds to the RG scale in the Kondo problem, so $g_{\text{eff}}(\theta) \equiv g e^{\epsilon\theta}$ signals the deformation from an operator of dimension $1 - \frac{\epsilon}{2}$. On the other hand, at the UV fixed points, $\sigma_+ J^+$ and $\sigma_- J^-$ have deformed dimensions $h = k\delta^2 - 2\delta + 1$. We, therefore, conjecture the following identification

$$1 - \frac{\epsilon}{2} = k\delta^2 - 2\delta + 1. \tag{75}$$

4. The RG flow is produced by a $\lambda(\sigma_+ J^+ + \sigma_- J^-)$ deformation of the UV fixed point. We then expect $\lambda$ should just be $g$, upon an appropriate change of coordinate.

We will now solve the ODE

$$\psi'' = \left[ e^{2\theta} e^{2x}(1 - g e^{-\epsilon x})^k + \widetilde{\alpha}^2 \right] \psi, \quad \epsilon \in \left[ 0, \frac{2}{k} \right], \tag{76}$$

perturbatively in $g$, which will verify the conjectured correspondence above. The recipe for such perturbative calculation is given in [15], which we briefly outline here.

We start by writing down the perturbative solution in small $g$, more precisely in the region $g e^{-\epsilon x} < 1$. Plug in $\psi_{II} = \sum g^i \psi^{(i)}$. At the leading order of $g$, we have

$$\partial_x^2 \psi^{(0)} - \left( e^{2\theta + 2x} + \widetilde{\alpha}^2 \right) \psi^{(0)} = 0, \tag{77}$$

$$\partial_x^2 \psi^{(1)} - \left( e^{2\theta + 2x} + \widetilde{\alpha}^2 \right) \psi^{(1)} = -e^{-x\epsilon} e^{2\theta + 2x} k \psi^{(0)}. \tag{78}$$

The solutions are

$$\psi^{(0)} = c_0 K_{\widetilde{\alpha}} \left( e^{\theta + x} \right), \tag{79}$$

$$\psi^{(1)} = c_0 k \left[ K_{\widetilde{\alpha}} \left( e^{x+\theta} \right) \int_?^x dx' \, I_{\widetilde{\alpha}}(e^{x'+\theta}) K_{\widetilde{\alpha}} \left( e^{x'+\theta} \right) e^{-\epsilon x'} e^{2\theta + 2x'} \right.$$

$$\left. + I_{\widetilde{\alpha}} \left( e^{x+\theta} \right) \int_x^{+\infty} dx' \, K_{\widetilde{\alpha}} \left( e^{x'+\theta} \right)^2 e^{-\epsilon x'} e^{2\theta + 2x'} \right] + c_1 \psi^{(0)}, \tag{80}$$

for arbitrary constant $c_0$ and $c_1$. Note that the lower limit of the first integration is arbitrary since $c_1$ is arbitrary.

We need to match with the asymptotic behaviours of the Bessel functions in the other region $g e^{-\epsilon x} > 1$, or roughly speaking large negative $x$. With the assumption $\epsilon \in (0, \frac{2}{k})$, the equation takes the simple form

$$\psi_I'' = \widetilde{\alpha}^2 \psi_I, \tag{81}$$

whose solutions are parametrized by[10]

$$\psi_I[g, \theta] = -Q[g_{\text{eff}}] g^{\alpha/\epsilon} \frac{1}{\sqrt{2\alpha}} e^{-\alpha x} - \widetilde{Q}[g_{\text{eff}}] g^{-\alpha/\epsilon} \frac{1}{\sqrt{2\alpha}} e^{\alpha x}. \tag{83}$$

---

[10]Since

$$K_{\widetilde{\alpha}}(x) = \frac{\pi}{2} \frac{I_{-\widetilde{\alpha}}(x) - I_{\widetilde{\alpha}}(x)}{\sin \widetilde{\alpha}\pi}, \tag{82}$$

to compare the asymptotics, without loss of generality here in this section we assume $\widetilde{\alpha} \geq 0$. The other sign works the same way and the final result is invariant under $\widetilde{\alpha} \to -\widetilde{\alpha}$ since the original ODE (76) is.

We can then consider the solution $\psi_{II}$ in the limit $x \to -\infty$, and use the asymptotics of the Bessel functions

$$K_{\widetilde{\alpha}}(e^x) \sim 2^{-1-\widetilde{\alpha}}e^{\widetilde{\alpha}x}\Gamma[-\widetilde{\alpha}] + 2^{-1+\widetilde{\alpha}}e^{-\widetilde{\alpha}x}\Gamma[\widetilde{\alpha}]\,, \tag{84}$$

$$I_{\widetilde{\alpha}}(e^x) \sim \frac{2^{-\widetilde{\alpha}}}{\Gamma[\widetilde{\alpha}+1]}e^{\widetilde{\alpha}x}\,, \tag{85}$$

we find

$$\begin{aligned}
\psi_{II} &\approx \psi^{(0)} + g\psi^{(1)}\,, \\
&\sim c_0\left(2^{-1-\widetilde{\alpha}}e^{\widetilde{\alpha}(x+\theta)}\Gamma[-\widetilde{\alpha}] + 2^{-1+\widetilde{\alpha}}e^{-\widetilde{\alpha}(x+\theta)}\Gamma[\widetilde{\alpha}]\right) \\
&\quad + c_0 g e^{\epsilon\theta}k\left[\left(2^{-1-\widetilde{\alpha}}e^{\widetilde{\alpha}(x+\theta)}\Gamma[-\widetilde{\alpha}] + 2^{-1+\widetilde{\alpha}}e^{-\widetilde{\alpha}(x+\theta)}\Gamma[\widetilde{\alpha}]\right)D + \left(\frac{2^{-\widetilde{\alpha}}}{\Gamma[\widetilde{\alpha}+1]}e^{\widetilde{\alpha}(x+\theta)}\right)C(\epsilon,\widetilde{\alpha})\right]\,,
\end{aligned}$$

where

$$C(\epsilon,\widetilde{\alpha}) := \int_{-\infty}^{+\infty}e^{(2-\epsilon)x'}K_{\widetilde{\alpha}}\left(e^{x'}\right)^2 dx' = \frac{\sqrt{\pi}}{4}\frac{\Gamma\left[1-\frac{\epsilon}{2}\right]}{\Gamma\left[\frac{3}{2}-\frac{\epsilon}{2}\right]}\Gamma\left[1-\frac{\epsilon}{2}-\widetilde{\alpha}\right]\Gamma\left[1-\frac{\epsilon}{2}+\widetilde{\alpha}\right]\,, \tag{86}$$

and $D$ is arbitrary due to the arbitrary constant $c_1$ in the solution (80). One natural choice is to fix $T_1(\theta,\epsilon,\widetilde{\alpha})=1$ identically by choosing

$$D = -\frac{\sin\frac{\pi}{2}(2\widetilde{\alpha}+\epsilon)}{\pi\cos\frac{\pi\epsilon}{2}}C(\epsilon,\widetilde{\alpha})\,, \tag{87}$$

which give us

$$T_2 = 2\cos\pi\widetilde{\alpha} - gke^{\epsilon\theta}\frac{1}{2\sqrt{\pi}}\tan\frac{\pi\epsilon}{2}\frac{\Gamma\left[1-\frac{\epsilon}{2}\right]}{\Gamma\left[\frac{3}{2}-\frac{\epsilon}{2}\right]}\Gamma\left[1-\frac{\epsilon}{2}-\widetilde{\alpha}\right]\Gamma\left[1-\frac{\epsilon}{2}+\widetilde{\alpha}\right](\cos 2\pi\widetilde{\alpha}-\cos\pi\epsilon)\,.$$

So upon identifying this $\widetilde{\alpha}$ with the one defined in (43) in the previous section and

$$k\delta^2 - 2\delta + 1 = \frac{\widetilde{\beta}^2}{4} \longleftrightarrow 1 - \frac{\epsilon}{2}\,, \tag{88}$$

$$\lambda^2 \longleftrightarrow -g\,, \tag{89}$$

we find an exact match with (44)!

A remark is in order. In writing down the ODE (52), we required $\epsilon \in (0, \frac{2}{k})$ to have desired behaviours around the negative infinity. What does such a requirement mean in the Kondo problem? Since the dimension of the defect changing operators $\left[t^{\mp}\mathcal{J}^{\pm}\right]_{\delta}$ in the UV is $h = 1-\frac{\epsilon}{2}$ and we require them to be relevant, i.e. $h < 1$, we might naively conclude a different range $\epsilon \in (0,2)$. The discrepancy is resolved by noticing $h = k\delta^2 - 2\delta + 1$ actually has a minimum $1-\frac{1}{k}$ at $\delta = \frac{1}{k}$. So in terms of $\epsilon$, we indeed have $\epsilon \in (0,\frac{2}{k})$, which serves as an amusing check.

## 3.3 Infrared analysis

### 3.3.1 IR physics

Let's first recall the key aspects of the isotropic Kondo lines. The global symmetry[11] of the isotropic Kondo model is $SU(2)$, The analysis of the IR physics is then two-folded. Firstly, we

---

[11]Technically we cannot discuss the symmetry group without specifying the information of the anti-chiral part of the bulk CFT, which we try to avoid. Therefore, the *symmetry* we consider here is only about the operator algebra, which might have 't Hooft anomaly or other global issues when we realize the symmetry on the Hilbert space.

would like to know the properties of the defect lines at the IR fixed point. Secondly, we would like to know what are the corrections to the physical observables, e.g. expectation value of the defect lines away from the IR fixed point. This is done by looking for $SU(2)$ invariant irrelevant operators in the IR. It has been known for a long time [34,35] that the answer depends on how the dimension $n = 2j + 1$ of the representation of the spin is compared to the level $k$.

- $j < \frac{k}{2}$: the Kondo defect line flowing to the IR fixed point becomes the Verlinde line with label $j$ and the corrections come from the descendant of the spin 1 primary $\mathcal{J}^a_{-1}\phi^a$ with dimension $\frac{2}{k+2} + 1$.

- $j \geq \frac{k}{2}$: the Kondo defect line flowing to the IR fixed point becomes the tensor product between the invertible Verlinde line of label $k/2$ and an IR-free Kondo defect line with an impurity of spin $j - \frac{k}{2}$. The corrections come from the operator $\mathcal{J}^a\mathcal{J}^a$ of dimension 2.

Let's now discuss the global symmetry in the anisotropic Kondo problem. Since we break the reletive coefficient between $[\mathcal{J}^+t^- + \mathcal{J}^-t^+]_\delta$ and $\mathcal{J}^0 t^0$, we are left with automorphism group $O(2) = SO(2) \rtimes \mathbb{Z}_2$. The action of $SO(2) = U(1)$ is

$$t^\pm \mapsto e^{\pm i\theta} t^\pm, \quad t^0 \mapsto t^0, \tag{90}$$

whereas $\mathbb{Z}_2$ acts by

$$t^+ \mapsto t^-, \quad t^- \mapsto t^+, \quad t^0 \mapsto -t^0. \tag{91}$$

One can indeed verify together they form the group[12] $O(2)$.

In the case of spin $j = \frac{1}{2}$ or $n = 2$, below are some examples of possible IR scenarios with the leading $O(2)$-preserving irrelevant deformations.

The IR defect could be the same as the isotropic case, namely a Verlinde line $\mathcal{L}_{\frac{1}{2}}$ of spin $j = \frac{1}{2}$ for the current algebra $\widehat{su(2)}_k$

- At $k = 1$, the only operators living on the line $\mathcal{L}_{\frac{1}{2}}$ are current algebra descendant of the identity operator, starting from $\mathcal{J}^0\mathcal{J}^0$, $\mathcal{J}^+\mathcal{J}^-$ and $\mathcal{J}^-\mathcal{J}^+$ with dimension 2.

- At $k \geq 2$, line $\mathcal{L}_{\frac{1}{2}}$ support a spin 1 primary operator denoted as $\phi^a$. So the leading $O(2)$-invariant operators are $\mathcal{J}^0_{-1}\phi^0$, $\mathcal{J}^+_{-1}\phi^-$ and $\mathcal{J}^-_{-1}\phi^+$ of dimension $1 + \frac{2}{k+2}$, which is small than that of $\mathcal{J}^0\mathcal{J}^0$, $\mathcal{J}^+\mathcal{J}^-$ and $\mathcal{J}^-\mathcal{J}^+$.

The IR line defect might be Verlinde lines for the chiral algebra[13] $\widehat{u(1)}_k$, which is the current algebra from $i\partial\varphi$ and extended by the vertex operator $: e^{i2\sqrt{k}\varphi}:$ of dimension $k$. There are $k$ Verlinde lines $\mathcal{L}^{u(1)}_s$, labelled by $s = 0, \ldots k - 1$, all of which have quantum dimension 1.

Another natural possibility is having an IR-free Kondo defect line of spin $j = \frac{1}{2}$, which becomes a direct sum of two identity lines with a possible twist $\delta_{IR}\mathcal{J}^0 t^0$ in the far IR. This mirrors what happens in the UV: There are operators of the form $[J^+t^-]_{\delta_{IR}} + [J^-t^+]_{\delta_{IR}}$ with the dimension derived in (33), $h = k\delta_{IR}^2 - 2\delta_{IR} + 1$. Recall in the UV, as discussed in section 2, we use them to initiate a relevant RG flow with $\delta_{UV} \in [0, \frac{2}{k}]$. They will be natural irrelevant operators in the IR with $\delta_{IR}$ taken outside this region.

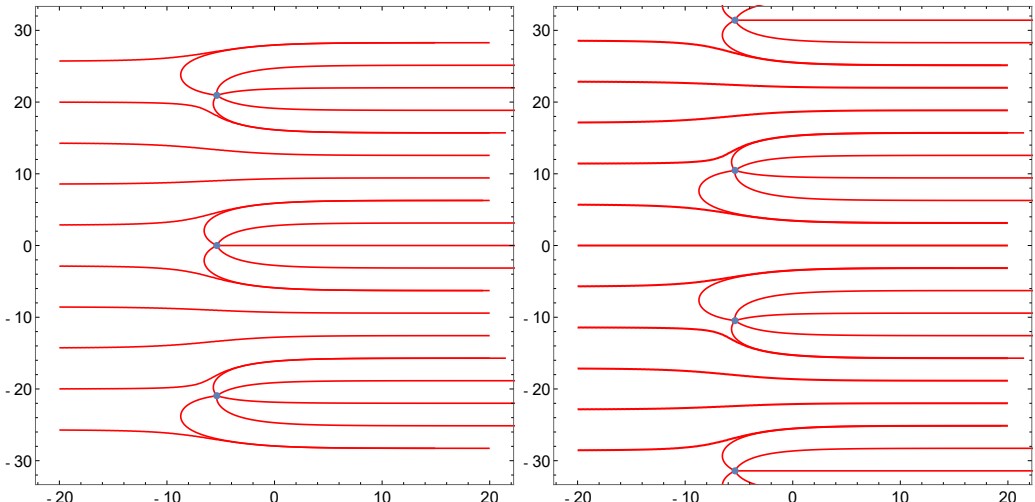

Figure 4: WKB diagram for $\epsilon = \frac{3}{10}$, $g = \frac{1}{5}$ and $k = 3$ on the left and $\epsilon = \frac{3}{10}$, $g = -\frac{1}{5}$ and $k = 3$ on the right.

### 3.3.2 WKB analysis in the IR

In this section, we perform the exact WKB analysis on the ODE (52) in the limit $\theta \to \infty$, which can tell us the infrared behaviours of the line defect. Here we are using a more refined WKB analysis developed in [15, 16] than the Voros/GMN-style one since the latter is only applicable to meromorphic potentials with simple zeroes [64–68]. We will refer the readers to the appendices of [15, 16] and references therein for more details.

**Physical RG flow with** $g > 0$: this corresponds to considering WKB analysis in large real $\theta$ limit for the ODE (52) with $g > 0$.

An example of the WKB diagrams is given in the left panel of figure 4. There is an infinite sequence of zeros of order $k$. Local behaviours around each zero are exactly the same as the order $k$ zero considered in the isotropic case. Since we are only interested in the case of $j = \frac{1}{2}$, i.e. $n = 2$

$$T_2[\theta] \equiv i\left(\psi_0(x; \theta - i\pi), \psi_0(x, \theta + i\pi)\right) = i\left(\psi_{-1}(x; \theta), \psi_1(x; \theta)\right), \tag{92}$$

whose associated Stokes lines are connected at the zero on the real axis, we will only need to analyze the local behaviour around this zero. So we have the same behaviour as in the isotropic case: the leading order of $T$-functions is the quantum dimension of the $\widehat{su(2)}_k$ current algebra. And the corrections come in powers of $\gamma = e^{-\theta \frac{2}{k+2}}$. This means that in this region of the parameter, the anisotropic Kondo lines flow to Verlinde lines $\mathcal{L}_{\frac{1}{2}}$ as in the isotropic case. But this makes sense since there is no $O(2)$- preserving operators that have dimension smaller than $1 + \frac{2}{k+2}$. This provides a direct derivation of the conjectured result from an infinitesimal analysis in small $\epsilon$ given in [43], whereas for us $\epsilon$ can be finite.

**'Unphysical' RG flow with** $g < 0$: this corresponds to considering WKB analysis in large real $\theta$ limit for the ODE (52) with $g < 0$. Given (89), this RG flow might look 'unphysical' since it will correspond to the coupling $\lambda$ being purely imaginary in the Kondo line defect (34). However,

---

[12]This actually cannot be true since $O(2)$ is not a subgroup of $SU(2)$. We conjecture the correct symmetry group is its double cover Pin$_-(2)$, which is a subgroup of $SU(2)$. In other words, there is a 't Hooft anomaly, which we should be able to detect by considering the action of the operator algebra on the Hilbert space. We will leave this as a future direction.

[13]Our normalization of $k$ is such that $\widehat{su(2)}_1 \cong \widehat{u(1)}_1$ as chiral algebras and the $\mathbb{Z}_k$ parafermion is given by the coset $su(2)_k/u(1)_k$.

surprisingly it is physically meaningful to consider such an analytically continued line defect including complex scale $\theta$ and complex coupling $g$ in both formal theory and condensed matter applications [15, 69].

The WKB diagram is given in the right panel of figure 4. The structure is very similar to the case above with $g > 0$, except that two Stokes lines closest to the real axis (corresponding to $\psi_{-1}$ and $\psi_1$ respectively) are connected to the negative infinity instead of at a zero.

To analyze the local behaviour around the negative infinity, suppose $y = x - x_{-\infty}$ is the local coordinate around $x_{-\infty}$, whose real part is very large negative. In this coordinate, the potential is

$$e^{2\theta + 2x_{-\infty}} e^{2y} \left(1 - g e^{-\epsilon x_{-\infty}} e^{-\epsilon y}\right)^k. \tag{93}$$

We might want to choose

$$(2 - k\epsilon) x_{-\infty} + 2\theta + k \log(-g) = 0, \tag{94}$$

then the potential takes the form of

$$e^{2y} \left(e^{-\epsilon y} - g_{\text{eff}}(\theta)^{-1 - \frac{\epsilon k}{2 - k\epsilon}}\right)^k + \tilde{\alpha}^2, \tag{95}$$

with again $g_{\text{eff}}(\theta) \equiv g e^{\epsilon \theta}$. Since we assume $0 < \epsilon < \frac{2}{k}$, the exponent of $g_{\text{eff}}(\theta)$ is necessarily negative. In the infrared $g_{\text{eff}}(\theta) \to +\infty$, the leading potential is then $e^{(2 - k\epsilon)y} + \tilde{\alpha}^2$ and with appropriate normalization, we have the usual Wronskians

$$i(\psi_m, \psi_{m+n}) \sim n + \dots \tag{96}$$

The corrections come in integer powers of $g_{\text{eff}}(\theta)^{-1 - \frac{\epsilon k}{2 - k\epsilon}}$, which, by dimensional analysis, corresponds to an irrelevant operator with the dimension

$$h = \frac{\epsilon}{2} \left(1 + \frac{\epsilon k}{2 - k\epsilon}\right) + 1. \tag{97}$$

It must be a boundary-changing operator associated with a direct sum of two identity lines with deformation $\delta_{\text{IR}} t^0 J^0$. In section 2, we have derived its scaling dimension to be $h_{\text{IR}} = k\delta_{\text{IR}}^2 - 2\delta_{\text{IR}} + 1$. Equating with (97), we find

$$\delta_{\text{IR}} = \frac{1}{k} \left(1 - \frac{1}{\sqrt{1 - \frac{k\epsilon}{2}}}\right) = \frac{1}{k} \left(1 - \frac{1}{|1 - k\delta_{\text{UV}}|}\right), \tag{98}$$

where $\delta_{\text{UV}}$ is what we call $\delta$ in section 2, i.e. the deformation parameter labelling the UV fixed point. We add this subscript to emphasize the difference from $\delta_{\text{IR}}$.

**When $\epsilon$ becomes larger**

When $\epsilon$ is small, zeros are far apart. Each zero source $k + 2$ Stokes lines and we are approximately dealing with isotropic case close to each zero. This has been made precise in section 3.1. When $\epsilon$ becomes large, zeros come closer but the span in the imaginary direction of the $k + 2$ lines stays the same. We might worry that the span of $k + 2$ lines between neighbouring zeros would overlap. Indeed, zeros of $1 - g e^{\epsilon x} = 0$ for $g > 0$ are given by

$$x_0 = \frac{\log g + 2\pi i n}{\epsilon}, \quad n \in \mathbb{Z}. \tag{99}$$

So zeros are separated by $\frac{2\pi i}{\epsilon}$ in the imaginary direction. Since each zero sources $k + 2$ WKB lines that span $(k + 1)\pi$ in the imaginary direction, they will overlap when $\frac{2}{k+1} < \epsilon < \frac{2}{k}$. This is particularly relevant when $k$ is small. We plot some WKB diagrams for $k = 3$ and $g > 0$

in figure 5. When $j < \frac{k}{2}$, the behaviour of physical RG flow (real $\theta$) stays the same, i.e. it flows to the Verlinde line $\mathcal{L}_j$ of $\widehat{su(2)}_k$ as in the isotropic case. However, the physical RG flow for $j > \frac{k}{2}$ and the complex RG flow exhibit an increasing complexity as the WKB diagram gets more complex when $\epsilon$ is larger. None of this can be deduced from the Kondo defect picture. This perfectly exemplifies the advantage of ODE/IM correspondence and exact WKB analysis in deriving the IR behaviours of the defect RG flows. When $\epsilon > \frac{2}{k}$, there will also be *small solutions* associated with the negative infinity, rather than just positive infinity, which is outside the scope of this paper.

# 4 Generalizations and future directions

## 4.1 Multichannel generalization

We have verified that the Stokes data of the proposed ODE (52) matches the expectation value of the anisotropic Kondo defect line in the chiral $SU(2)_k$ WZW model. In the construction of 4d Chern Simons, the generalization to the multichannel case $\prod_i SU(2)_{k_i}$ is obvious. We just need to have multiple insertions of surface defect wrapping $\mathbb{R}^2 \times \{z_1, z_2 \ldots\}$. The meromorphic one-form is

$$\frac{1}{\hbar} \frac{dz}{z} \sum_{i=1} \left(1 + \frac{k}{2} \frac{\hbar}{z/z_i - 1}\right) \equiv dx \sum_i \left(1 + \frac{k}{2} \frac{\epsilon g_i}{e^{\epsilon x} - g_i}\right), \tag{100}$$

where $z = e^{\epsilon x}$, $\epsilon = \hbar$, and $z_i = g_i$. Therefore the ODE is given by

$$\partial_x^2 \psi(x) = \left(e^{2\theta} e^{2x} \prod_i \left(1 - g_i e^{-\epsilon x}\right)^{k_i} + t(x)\right) \psi(x), \tag{101}$$

where $t(x) \equiv a_+^2 + \partial a_+$ with the straightforward generalization of $a_+(x)$

$$a_+(x) = -\alpha_+ - \sum_i \epsilon l_i \frac{g_i}{e^{\epsilon x} - g_i} + \epsilon \sum_a \frac{e^{\epsilon u_a}}{e^{\epsilon x} - e^{\epsilon u_a}} - \epsilon \sum_b \frac{e^{\epsilon u_b'}}{e^{\epsilon x} - e^{\epsilon u_b'}}, \tag{102}$$

where $u_a$ and $u_b'$ are fixed by the trivial monodromy condition as in (66) and (67). As for the Kondo defect, we just replace $\lambda \left[t^\pm \mathcal{J}^\mp\right]_\delta$ with $\sum_i \lambda_i \left[t^\pm \mathcal{J}_i^\mp\right]_\delta$, which can be checked with the ODE (101) above by a straightforward but a bit tedious calculation in the fashion of the appendix of [16]. Since the structure is the same, we will not repeat it here. Note that one might be tempted to turn on different $\delta_i$ for different factors of $SU(2)$, which corresponds to having $\epsilon_i$ different from each other. However, there is no obvious way to generalize (102) or to generalize the 4d Chern Simons theory to have multiple $\hbar_i$. Therefore we conjecture such line defects are not integrable. It would be interesting to explore this further.

## 4.2 Higher spin

In this article, one of the pieces of evidence for the ODE/IM correspondence is that we have verified explicitly (68) in the UV for $n = 2$ or spin $\frac{1}{2}$. While the construction of $T_n[\theta]$ as Stokes data from the ODE side works for any $n$, we only discussed the anisotropic Kondo defect for $n = 2$. In the case with impurities of higher spin $n > 2$, the main idea is the same: the UV fixed point we start from is a certain marginal deformation of a direct sum of $n$ identity defect; then the RG flow is initiated by turning on defect-changing operators. However, both the space of marginal deformations and the space of defect-changing operators are much larger and the unbroken $U(1)$ symmetry does not give us enough constraints as it does for $n = 2$. A

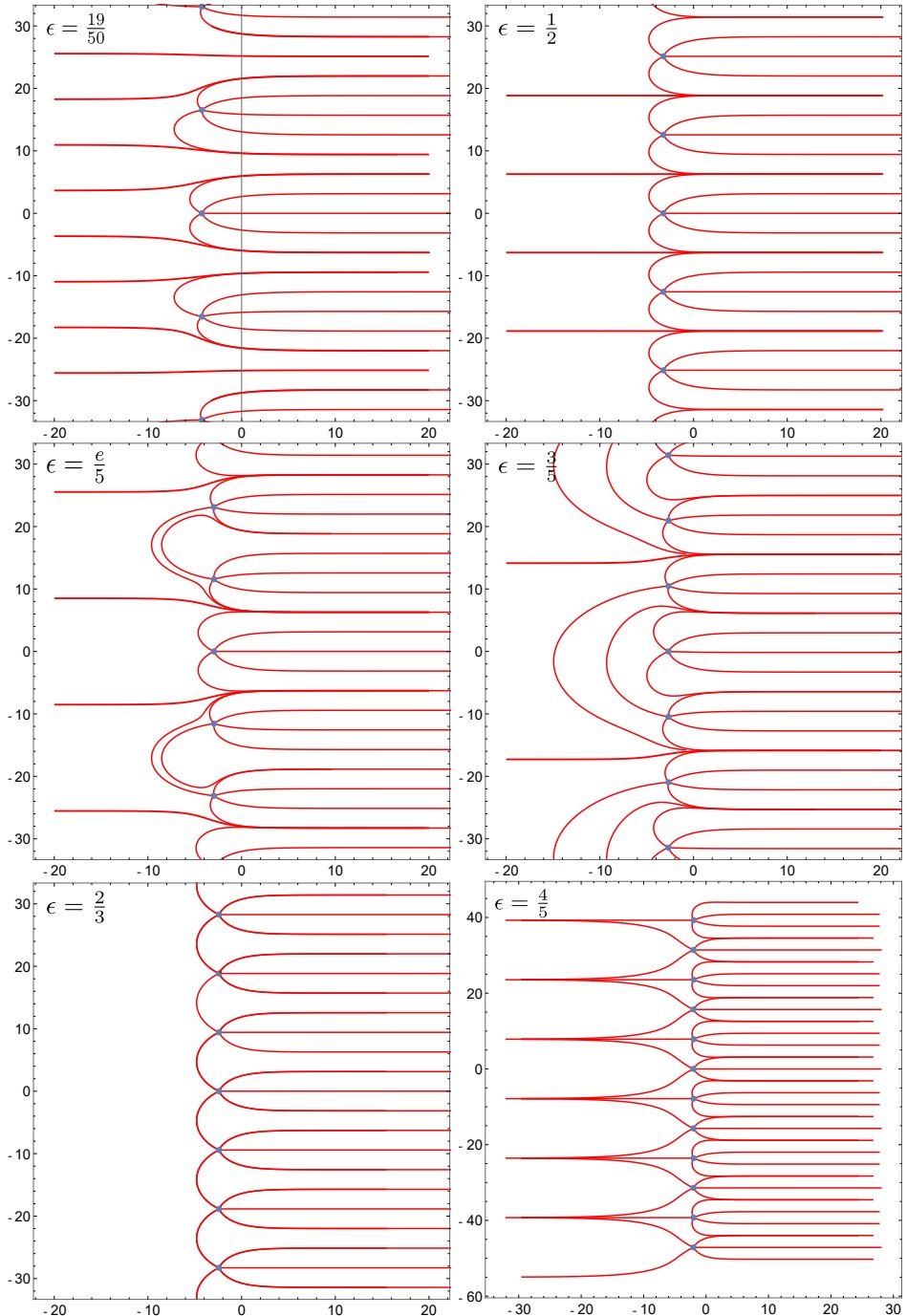

Figure 5: WKB diagrams for $k = 3$, $g = \frac{1}{5}$ and a sequence of increasing $\epsilon$ as labelled inside the figures. The zeros cease to be isolated when the span of $k+2$ lines overlap, i.e. when $\epsilon \in \left[\frac{2}{k+1}, \frac{2}{k}\right]$. When $\epsilon > \frac{2}{k}$, there will also be small solutions associated with the negative infinity rather than just positive infinity, which is outside the scope of this paper. Whenever needed (except $\epsilon = \frac{2}{3}$), a small imaginary part is given to the potential $P(x)$ to avoid the appearance of the saddle connection, i.e. Stokes line connecting two zeros. See the discussion of $\vartheta$-WKB diagram in [68] for more details.

priori, we don't know which RG flow in this large space of couplings is integrable. Based on our experience with 4d Chern Simons theory in the trigonometric setting [37, 38], we expect to find the finite-dimensional matrix representations[14] of $U_q(sl_2)$, which in principle can be verified by enforcing the commutativity relation (3) and the Hirota fusion relation (4).

In fact, due to the simplicity[15] of the renormalization as we discussed at the end of the section 2, we can explicitly show the equivalence of the anisotropic Kondo defect with the transfer matrix [18, 54, 70] for $U_q(\widehat{sl(2)})$. Using the notation used in this article, the transfer matrix for the level $k = 1$ reads[16,17]

$$\mathbf{T}_\ell(\lambda) \equiv \text{Tr}_\ell \left[ e^{i2\pi\beta t^0 J_0^0} \text{Pexp}\left( \widetilde{\lambda} \int_0^{2\pi} du \left( V_{-\beta} q^{t^0} t^+ + V_\beta q^{-t^0} t^- \right) \right) \right] \tag{103}$$

$$= \text{Tr}_\ell \left[ e^{i2\pi\beta t^0 J_0^0} \sum_{m=0}^\infty \widetilde{\lambda}^m \sum_{\sigma_1,\ldots,\sigma_m=\pm} \left( q^{\sigma_1 t^0} t^{\sigma_1} \right) \ldots \left( q^{\sigma_m t^0} t^{\sigma_m} \right) \right. \tag{104}$$

$$\left. \times \int_{2\pi > u_1 > \cdots > u_m > 0} du_1 \ldots du_m V_{-\sigma_1\beta}(u_1) \ldots V_{-\sigma_m\beta}(u_m) \right], \tag{105}$$

where $t^0$, $t^\pm$ are generators of $U_q(sl_2)$ and the trace is taken in the $2l + 1$ dimensional matrix representation. The deformation parameter $q = e^{i\pi\beta^2/4}$

Let's first look at the $\ell = \frac{1}{2}$ case. Note that for the two-dimensional representation, we have

$$\left( q^{\pm t^0} t^\pm \right)^2 = q^{-1} q^{2t^0} \left( t^\pm \right)^2 = 0, \tag{106}$$

$$q^{t^0} t^+ q^{-t^0} t^- = q t^+ t^-, \tag{107}$$

$$q^{-t^0} t^- q^{t^0} t^+ = q t^- t^+, \tag{108}$$

we then find

$$\mathbf{T}_{\frac{1}{2}}(\lambda) = \text{Tr}_\ell \left[ e^{i2\pi\beta t^0 J_0^0} \sum_{m=0}^\infty \widetilde{\lambda}^{2m} q^m \left( t^+ t^- \right)^m \right.$$
$$\times \int_{2\pi > u_1 > \cdots > u_{2m} > 0} du_1 \ldots du_{2m} V_{-\beta}(u_1) V_{+\beta}(u_2) \ldots V_{-\beta}(u_{2m-1}) V_{+\beta}(u_{2m}) \right]$$
$$+ \text{Tr}_\ell \left[ e^{i2\pi\beta t^0 J_0^0} \sum_{m=0}^\infty \widetilde{\lambda}^{2m} q^m \left( t^- t^+ \right)^m \right.$$
$$\left. \times \int_{2\pi > u_1 > \cdots > u_{2m} > 0} du_1 \ldots du_{2m} V_{+\beta}(u_1) V_{-\beta}(u_2) \ldots V_{+\beta}(u_{2m-1}) V_{-\beta}(u_{2m}) \right]. \tag{109}$$

Using the identity used in (37) and the argument in figure 3, we can show that the twist $e^{4\pi b t^0 J_0^0}$ in the trace enables us to cyclically permute operators without introducing monodromies. Therefore two terms are actually the same. Recall from (20) that $J_0^0$ acts on a

---

[14]Note that $t^\pm$ and $t^0$ are precisely the two-dimensional representation of $U_q(sl_2)$.

[15]In contrast, we were not able to do this in [15] since the renormalization is very nontrivial.

[16]See also [52] whose convention we follow closely here. In particular $h \leftrightarrow 2t^0$, $e_\pm \leftrightarrow t^\pm$, $2\beta \leftrightarrow \beta$, $h_0 \leftrightarrow \frac{4}{i\beta} J_0^0$.

[17]When $k > 1$, the tranfer matrix is given in [22] using parafermion CFT. However, as we have seen in this article, it is a lot easier to embed $su(2)_k \subset (su(2)_1)^k$ since the Kondo defect won't feel the difference between these two. Therefore, we will only discuss $k = 1$ and $k > 1$ case just needs a trivial product of $k$ copies.

state[18] in the defect Hilbert space by

$$J_0^0 |\alpha\rangle = \frac{\alpha}{2}|\alpha\rangle, \tag{110}$$

we then find an exact match with the Kondo defect given in section 2 with the following identification

$$\lambda \leftrightarrow q^{\frac{1}{2}}\widetilde{\lambda}, \tag{111}$$

as before we can define $\widetilde{\alpha} \equiv \frac{1}{2}\beta\alpha$. The equivalence we just derived between the spin $\frac{1}{2}$ anisotropic Kondo defect and the transfer matrix for $U_q(sl_2)$ confirms the common lore: the anisotropic Kondo line defect is integrable when the matrix $t^0$, $t^\pm$ are in the representations of $U_q(sl_2)$ instead of $sl_2$. We will leave it as a future direction to explicitly verify this in the higher spin.

## 4.3 Anisotropic vs coset

From the ODE (101), it is interesting to perform a change of coordinate $y = e^{-\epsilon x}$ and we find

$$e^{2\theta}\epsilon^{-2}y^{-(2+\frac{2}{\epsilon})}\prod_i(1 - g_i y)^{k_i} + \tilde{t}(y), \tag{112}$$

where $\tilde{t}(y) \equiv \epsilon^{-2}y^{-2}t(x) - \frac{1}{4y^2}$. This is precisely the ODE proposed in equation (7.7) of [52]. On the other hand, according to [15], it describes a Kondo defect in a coset

$$\frac{\mathfrak{su}(2)_{-(2+\frac{2}{\epsilon})} \oplus \prod_i \mathfrak{su}(2)_{k_i}}{\mathfrak{su}(2)_{-(2+\frac{2}{\epsilon})+\sum k_i}}, \tag{113}$$

where the excited states are identified with those of the (101) after a spectral flow due to the Schwarzian contribution $-\frac{1}{4y^2}$. The existence of the branch cut makes it less preferred for the analysis than (101). However, abstractly it would be really interesting to understand the relationship between two interpretations. This curiosity already exists in the isotropic ODE (5), but it seems to be a bit more intriguing in the anisotropic case. For example, we can try to lift them to 4D Chern Simons theory. They both have a 4d Chern Simons construction in a trigonometric setting on $\mathbb{C}^*$ but with different boundary conditions at the origin.

# Acknowledgments

The author thanks Davide Gaiotto for suggesting the original idea, participation in the early stages of the project and countless in-depth discussions. The author also would like to thank Benoit Vicedo for inspiring discussions. The author is grateful to Gleb A. Kotousov and Sergei L. Lukyanov for sharing a draft of their unpublished work.

**Funding information** This research is supported in part by a grant from the Krembil Foundation by the Perimeter Institute for Theoretical Physics. Research at Perimeter Institute is supported in part by the Government of Canada through the Department of Innovation, Science and Economic Development Canada and by the Province of Ontario through the Ministry of Colleges and Universities.

---

[18]Again, writing $|\alpha\rangle$ is an abuse notation since one has to remember it is a state in the defect Hilbert space that corresponds via state/operator correspondence to the twist operator $V_\alpha$ living at the end of the twist line $\mathcal{L}_\alpha$, as we have carefully explained in section 2.

# A  Configuration space

The *configuration space of n ordered points*, $\text{Conf}_n(X)$ of a topological space $X$ is the space of $n$ distinct points in $X$.

$$\text{Conf}_n(X) := \{(x_1,\ldots,x_n)|x_i \in X, x_i \neq x_j, \forall i \neq j\} = X^n - \Delta, \tag{A.1}$$

where $\Delta$ is sometimes called *fat diagonal*, the space of $n$ points where at least two points coincide. The *unordered* configuration space of $n$ points is simply the quotient by the permutation group $\text{Conf}_n(X)/S_n$.

Let's demonstrate this in a few examples.

**Example 1.** Let $I$ be the open interval $I = (0,1)$. It is easy to see $\text{Conf}_n(I)/S_n$ is basically

$$\{(x_1,\ldots,x_n)|0 < x_1 < x_2 \cdots < x_n < 1\}, \tag{A.2}$$

which is just the open $n$-simplex. For example, when $n = 1,2,3$, we have $I$, an open triangle, and an open tetrahedron respectively. $\text{Conf}_n(I)$ is then a disjoint union of $n!$ copies of $n$-simplexes.

From this, we can obtain our main interest: the space of $n$ points on a circle with an orientation.

**Example 2.** $\text{Conf}_n(S^1)$ can be obtained as follows. We put the first point anywhere on the circle $x_1 \in S^1$. Break open the circle into an open interval $I$. Ways of putting the remaining $n-1$ points on the open interval is precisely $\text{Conf}_{n-1}(I)$. More formally speaking, there is a homeomorphism

$$f : S^1 \times \text{Conf}_{n-1}(I) \to \text{Conf}_n(S^1). \tag{A.3}$$

Therefore, $\text{Conf}_n(S^1)$ is just the product of a circle and a disjoint union of $(n-1)!$ open simplices.

In particular, $\text{Conf}_2(S^1)$ is just a cylinder, or equivalently a torus with diagonal removed.

# B  Lie algebra conventions

We follow the convention from [15]. Our normalization convention for the spin basis of $\mathfrak{sl}_2$ is

$$t^{\pm} = \frac{1}{\sqrt{2}}(t^1 \pm i t^2), \quad t^0 = \frac{1}{\sqrt{2}}t^3, \tag{B.1}$$

which satisfies the relations

$$[t^0, t^{\pm}] = \pm t^{\pm}, \quad [t^+, t^-] = 2t^0. \tag{B.2}$$

The relations in the corresponding untwisted affine Kac-Moody algebra $\widetilde{\mathfrak{sl}}_2$ read

$$\left[J_n^0, J_m^0\right] = \frac{\kappa n}{2}\delta_{n+m,0}, \tag{B.3}$$

$$\left[J_n^0, J_m^{\pm}\right] = \pm J_{n+m}^{\pm}, \tag{B.4}$$

$$\left[J_n^+, J_m^-\right] = 2J_{n+m}^0 + \kappa n \delta_{n+m,0}, \tag{B.5}$$

for $n, m \in \mathbb{Z}$. Let $|l, \kappa\rangle$ denote the ground state in the spin $l$ module at level $\kappa$.

Spectral flow [71, 72] is an automorphism of $\widehat{\mathfrak{sl}}_2$ given, for $\alpha \in \mathbb{R}$, by

$$U_{\alpha}: \quad J_n^+ \mapsto J_{n+\alpha}^+, \quad J_n^- \mapsto J_{n-\alpha}^-, \quad J_n^0 \mapsto J_n^0 + \frac{k}{2}\alpha\delta_{n,0}, \tag{B.6a}$$

$$L_0 \mapsto L_0 + \alpha J_0^0 + \frac{k}{4}\alpha^2. \tag{B.6b}$$



There is also an involutive automorphism induced by the Weyl group $W(\mathfrak{sl}_2) = \mathbb{Z}_2$

$$w_1: \quad J_n^+ \mapsto J_n^-, \quad J_n^- \mapsto J_n^+, \quad J_n^0 \mapsto -J_n^0, \tag{B.7}$$

which satisfy

$$U_\alpha U_{\alpha'} = U_{\alpha+\alpha'}, \quad U_0 = w_1^2 = 1, \quad U_\alpha w_1 = w_1 U_{-\alpha}. \tag{B.8}$$

We therefore have $\mathrm{Aut}(\widehat{\mathfrak{sl}}_2) = \mathbb{R} \rtimes \mathbb{Z}_2$. In particular, the even part is inner and corresponds to the affine Weyl group $W(\widehat{\mathfrak{sl}}_2) = (2\mathbb{Z}) \rtimes \mathbb{Z}_2$. Consequently, the induced action by $U_{2\mathbb{Z}}$ maps each integral highest weight representation into itself, whereas more general $U_\alpha$ maps between the (twisted) modules. For example,

$$U_1: j \mapsto \frac{k}{2} - j, \quad j = 0, \frac{1}{2}, 1, \dots, \frac{k}{2}. \tag{B.9}$$

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
