# Peer review of "Anisotropic Kondo line defect and ODE/IM correspondence"

_SciPost Physics, doi:SciPost Phys. 15, 248 (2023)_

## Round 2 · Referee Report · Anonymous (Referee 1) · 2023-8-2

Report

Dear Editor,

I have read through the article 2106.07792v2 that was submitted to SciPost Physics for publication. It discusses the ODE/IM correspondence (sometimes referred to as the ODE/IQFT correspondence) for the Kondo model and its generalizations. For the most part, the author studies the anisotropic Kondo defect associated with the SU(2)_k WZW model. As far as I can tell, the analysis follows the standard approach, while the proposed ODE has already appeared before in the literature. An important aspect, that adds value to the work, is the connection to 4D Chern Simons theory. The latter, in the context of the theory of integrable systems, is a relatively new development that may potentially yield an interesting perspective on the field. I believe the most important original result that is contained in the manuscript is a proposal for the ODE/IM correspondence for the multichannel Kondo model, where the differential equation is given by formula (4.2). It is a pity that this is not developed further in the paper.

In my opinion, the paper has solid results and I would like to recommend it for publication. It is appropriate to be published as is and I don't insist on any further changes. However, the style of referencing is not quite to my taste. For instance, following eq.(1.4) the works [18-22] are cited after the mention of the ODE/IM correspondence. Of these, refs. 18, 21 and 22 contain pioneering results, while 19 and 20 are developments of these papers. I would avoid mixing in a single citation all of these works as it gives the impression that they are of equal importance.

Also, there are a few minor typos. In particular, in eq. (1.3) the subscript for the first $\hat{T}$ should be $n$ and not $n'$.
  • validity: -
  • significance: -
  • originality: -
  • clarity: -
  • formatting: -
  • grammar: -

Author:  Jingxiang Wu  on 2023-10-21  [id 4055]

(in reply to Report 1 on 2023-08-02)

We thank the referee for their careful reading and the suggestions to the manuscript. We will implement the improvement in the resubmission.

---

## Round 2 · Referee Report · Anonymous (Referee 2) · 2023-9-18

Report

The paper studies the anisotropic Kondo line defects in products of chiral SU(2) WZW models by proposing equivalent ODEs in the perspective of the ODE/IM correspondence: eqs. (3.7) regarding the vacuum state, (3.17) and (4.2) regarding excited states. Since the paper was preceded by [53] (‘we became aware of [53] which has some overlap with our results…..for sharing their work with us before publishing.’), it would need a more precise, point by point, clarification of the overlap results and its own new achievements, for instance about the WKB analysis and other topics. A similar distinction is needed for what concerns the vacuum state (3.7) with respect to [24], although of even minor relevance. In any case, these would be minor additions.

Nevertheless, the idea of the CS embedding (towards a string theory description) of the anisotropic case is new and valuable (section 3 and 4), as it relies on gauge anomaly, RG and a relevant one form. The paper is well and clearly written. Therefore, in the end the paper should be published provided the differences with [53] and [24] are better highlighted.

Typos: 1) we will only the chiral half of a CFT 2) wellstudied 3) ‘While the manuscript is close to completion’: is——>was 4) Eq. (2.1)——>log (z-w) 5) Period before (3.8) to be deleted 6) Sentence starting by ‘After a simple coordinate transformation…’ 7) Liouvlle——>Liouville

  • validity: -
  • significance: -
  • originality: -
  • clarity: -
  • formatting: -
  • grammar: -

Author:  Jingxiang Wu  on 2023-10-21  [id 4054]

(in reply to Report 2 on 2023-09-18)
Category:
remark

We thank the referee for their careful reading and the suggestions for the manuscript. We will implement the improvement in the resubmission.

---

## Editorial Decision

published